

# Coupling a global glacier model to a global hydrological model prevents underestimation of glacier runoff

Pau Wiersma[1,2], Jerom Aerts[1], Harry Zekollari[3,4], Markus Hrachowitz[1], Niels Drost[5], Matthias Huss[6,7,8], Edwin H. Sutanudjaja[9], and Rolf Hut[1]

[1]Department of Water Management, Faculty of Civil Engineering and Geosciences, Delft University of Technology, Delft, the Netherlands
[2]Institute of Earth Surface Dynamics, Faculty of Geosciences and Environment, University of Lausanne, Lausanne, Switzerland
[3]Department of Geoscience and Remote Sensing, Faculty of Civil Engineering and Geosciences, Delft University of Technology, Delft, the Netherlands
[4]Laboratoire de Glaciologie, Université Libre de Bruxelles, Brussels, Belgium
[5]Netherlands eScience Center, Amsterdam, the Netherlands
[6]Laboratory of Hydraulics, Hydrology and Glaciology (VAW), ETH Zürich, Zürich, Switzerland
[7]Swiss Federal Institute for Forest, Snow and Landscape Research (WSL), Birmensdorf, Switzerland
[8]Department of Geosciences, University of Fribourg, Fribourg, Switzerland
[9]Department of Physical Geography, Faculty of Geosciences, Utrecht University, Utrecht, the Netherlands

**Correspondence:** Pau Wiersma (pau.wiersma@unil.ch), Rolf Hut (r.w.hut@tudelft.nl)

**Abstract.** Global hydrological models have become a valuable tool for a range of global impact studies related to water resources. However, glacier parameterization is often simplistic or non-existent in global hydrological models. By contrast, global glacier models do represent complex glacier dynamics and glacier evolution, and as such hold the promise of better resolving glacier runoff estimates. In this study, we test the hypothesis that coupling a global glacier model with a global hydrological model leads to a more realistic glacier representation and consequently improved runoff predictions in the global hydrological model. To this end, the Global Glacier Evolution Model (GloGEM) is coupled with the global hydrological model PCR-GLOBWB 2 using the eWaterCycle platform. For the period 2001-2012, the coupled model is evaluated against the uncoupled PCR-GLOBWB 2 in 25 large-scale (>50.000 km$^2$) glacierized basins. The coupled model produces higher runoff estimates across all basins and throughout the melt season. In summer, the runoff differences range from 0.07% for weakly glacier-influenced basins to 252% for strongly glacier-influenced basins. The difference can primarily be explained by PCR-GLOBWB 2 not accounting for glacier flow and glacier mass loss, thereby causing an underestimation of glacier runoff. The coupled model performs better in reproducing basin runoff observations mostly in strongly glacier-influenced basins, which is where the coupling has the most impact. This study underlines the importance of glacier representation in global hydrological models and demonstrates the potential of coupling a global hydrological model with a global glacier model for better glacier representation and runoff predictions in glacierized basins.



# 1 Introduction

1.9 billion people worldwide rely on glacial meltwater as part of their water resources (Immerzeel et al., 2020). Glaciers can act as a crucial multiannual buffer, particularly in regions prone to drought (Pritchard, 2019; Biemans et al., 2019). Yet, as glaciers have been strongly retreating (Hugonnet et al., 2021) and are projected to continue doing so throughout the 21st century (Edwards et al., 2021), their role in the water cycle will change. On the intra-annual scale, peak runoff will occur earlier in summer, while on the interannual scale glacier mass loss will cause an initial peak in glacier runoff, followed by a steady decline until a new equilibrium is reached (Jansson et al., 2003; Huss and Hock, 2018). In many basins throughout the world this 'peak water' already lies in the past (Huss and Hock, 2018), indicating that the shift from a glacial to a nival-pluvial regime is well underway. This will not only impact the water supply of millions of people, but also lead to an increased potential for natural hazards, hydro-political tension (Immerzeel et al., 2020) and instability of many ecosystems influenced by glacial meltwaters (Cauvy-Frauníe and Dangles, 2019).

To be able to account for the changing contribution of glaciers to daily runoff, many hydrological models applied to glacierized catchments include glacier parameterization schemes to form glacio-hydrological models (see van Tiel et al. (2020) for an overview). These models have been applied both at a small catchment scale (e.g. Huss et al., 2008; Ragettli et al., 2016) as well as at a regional or multiple catchment scale (e.g. Farinotti et al., 2012; Frans et al., 2018). Another approach involves the use of glacier geometry evolution estimates of an independent glacier model as forcing to a hydrological model, which has likewise been applied on local (Laurent et al., 2020; Hanus et al., 2021) to regional (Brunner et al., 2019) scales.

On a global scale, however, the integration of glacier processes in hydrological modeling is still lacking. Global hydrological models (GHMs) have gained popularity in recent years and have been used to study many different global issues including flood hazard (e.g. Do et al., 2020; Aerts et al., 2020), drought propagation (e.g. Gevaert et al., 2018) and ecological degradation (e.g. Barbarossa et al., 2021). Nonetheless, GHMs are reported to have an overly simplistic description of glacier dynamics (Van Dijk et al., 2014; Cáceres et al., 2020) and to mostly treat glaciers as non-glacierized terrain (Cáceres et al., 2020). The complex and dynamic contribution of glacier runoff to basin runoff is therefore expected to be captured only to a limited degree by GHMs. This has been shown to cause problems in the application of GHMs to glacierized basins (Scanlon et al., 2018; Müller Schmied et al., 2020).

Models dedicated to simulating glacier evolution on a global scale exist in the form of global glacier models (GGMs) (Hock et al., 2019; Marzeion et al., 2020). These models combine a surface mass balance model with a glacier geometry change model, which can range in complexity from a volume-area-length scaling model (Marzeion et al., 2012; Radić and Hock, 2014), to a mass-conserving retreat parameterization (Huss and Hock, 2015) to a prognostic ice dynamical model (Maussion et al., 2019; Zekollari et al., 2019). Although most GGMs are developed with the goal of simulating the mass balance and evolution of glaciers, some also produce glacier runoff as a model output (Hirabayashi et al., 2010; Bliss et al., 2014; Huss and Hock, 2018). This makes them suitable for coupling with GHMs, where glacier runoff can potentially be used as a direct input.

Several studies have investigated the global contribution of glaciers to streamflow on a coarse temporal resolution. Kaser et al. (2010) compared glacier runoff with the mean upstream precipitation at several elevations to estimate the contribution





of glacier runoff along the course of a multitude of large glacier-fed rivers. To a similar purpose, Schaner et al. (2012) used
a land-surface hydrological model combined with an energy-balance model. Huss and Hock (2018) compared the runoff of a
GGM to monthly average basin runoff observations to assess the changing contribution to basin-scale runoff and the timing of
intra- and inter-annual peak water. In another recent study, Cáceres et al. (2020) coupled a GGM with a GHM to assess the joint
contribution of glacial and non-glacial water storage anomalies to ocean mass change. However, to the best of our knowledge,

no study so far has investigated whether global runoff predictions can be improved through the coupling of GHMs and GGMs.

In this study, we test the hypothesis that the coupling of a GGM with a GHM can lead to a more realistic glacier representation
and consequently improved daily runoff estimates by GHMs in glacierized basins. To this end, the GGM GloGEM (Huss and
Hock, 2015) is coupled with the GHM PCR-GLOBWB 2 (Sutanudjaja et al., 2018). We evaluate the coupled model for 25
large glacierized basins (>50.000 km$^2$) in North and South America, Europe, Asia and New Zealand through a comparison

with the uncoupled GHM, which serves as a benchmark. Through this approach, we aim at identifying structural differences
in behavior between the two models, as well as determining which model is the most suited at reproducing the observed basin
runoff.

## 2   Methods and Data

### 2.1   Global hydrological model

For the global hydrological modelling we used PCR-GLOBWB 2 (PCRaster GLOBal Water Balance model: version 2.0)
(Sutanudjaja et al., 2018), which can be considered a representative example of a GHM in terms of snow and glacier modeling.
Compared to other GHMs, it has the advantage of a relatively high resolution (5 arcmin, or  10 km at the equator) and the
ability to integrate human water use. Details on the model can be found in Sutanudjaja et al. (2018) and Beek et al. (2011).
The four standard land cover types are tall natural vegetation, short natural vegetation, non-paddy irrigated crops and paddy

irrigated crops, and there is an option to include custom land cover types. For latitudes up to 60 degrees, PRC-GLOBWB 2
relies on the Digital Elevation Model (DEM) of HydroSHEDS (Lehner et al., 2008), while for latitudes over 60 degrees the
lower resolution HYDRO1K DEM of USGS is used (Verdin and Greenlee, 1998). The snow module of PCR-GLOBWB is
based on the HBV snow module (Bergstrom, 1995) and accounts for accumulation, melt and refreezing using a degree-day
method, but no redistribution (e.g. sliding of the snowpack, avalanches) to other grid cells is considered. Glaciers are effectively

treated as static rock masses, i.e. the DEMs reflect the glacier surface elevation but glacier flow, snow compression and ablation
are not resolved. For the sake of consistency with the GGM, ERA-Interim Reanalysis temperature and precipitation data (Dee
et al., 2011) are used as forcing. Of note is that PRC-GLOBWB 2 requires no subsequent calibration.

### 2.2   Global glacier model

The Global Glacier Evolution Model (GloGEM) was developed by Huss and Hock (2015), while the data we use here is from

a more recent study (Huss and Hock, 2018) that specifically focused on glacier runoff. The data consists of the runoff of



individual glaciers in the 56 glacierized drainage basins across five continents with an area of more than 50.000 km$^2$, a glacier area of more than 30 km$^2$ and an ice cover of more than 0.01% of the basin area. The glacier runoff is defined as the total amount of water originating from the glacierized area defined in the Randolph Glacier Inventory (RGI) (Pfeffer et al., 2014), and is kept constant, i.e. runoff from the areas that become ice-free throughout the simulation remains accounted for. GloGEM

was run for the period 1980-2100, but here we only consider the simulation results from 2000-2012. This time interval for the present analysis is given by the first inventory date of most of the RGI glacier outlines used in GloGEM (Pfeffer et al., 2014) and the last year for which ERA-Interim Reanalysis forcing data were used. After conversion to hydrological years, the considered date range thus becomes October (April) 2000 to September (March) 2012 for the Northern (Southern) hemisphere.

The glacier runoff data, which is resolved at the level of individual glaciers, was preprocessed to match the spatial and tem-

poral resolution of PCR-GLOBWB 2. This consisted of a conversion to raster data of the same resolution as PCR-GLOBWB 2 (5 arcmin) and consequently a resampling from monthly to daily resolution. The resampling was performed with a weighting function based on the ERA-Interim surface temperature data (Eqs 1 and 2). Only days with a daily mean temperature below $-5°$C were excluded from the weighting, since melt can still occur on days with a mean air temperature below 0 °C due to strong irradiation (Ayala et al., 2017) or a positive maximum temperature. Despite the existence of a strong day-night cycle

over glaciers, a resampling to diurnal resolution was not possible given the daily time step of PCR-GLOBWB 2.

$$w_D = \frac{1 + \alpha \cdot \frac{T_D - \overline{T_{T>268}}}{\overline{T_{T>268}}}}{N_{T>268}} \cdot (T_D > 268) \tag{1}$$

$$R_D = w_D \cdot R_M \tag{2}$$

Here, $w_D$ is the weight given to a particular day, $T_D$ is the mean daily surface temperature in Kelvin, $\overline{T_{T>268}}$ is the average of

all mean daily temperatures above 268 K in the considered month and $N_{T>268}$ is the number of days with a mean daily temperature above 268 K. $\alpha$ is a weighting factor that was set to 20 after calibration on the runoff of the Great Aletsch glacier (BAFU, 2020) (see Supplementary figure S1). A sensitivity analysis of $\alpha$ is given in section 6 of the Supplementary material. Finally, $R_D$ and $R_M$ are the daily and monthly grid cell glacier runoff respectively. This weighting function is mass-conserving, since it is linear in nature and the sum of the weights always equals $N_{T>268}$,


### 2.3   Model platform

We generated all model-specific ERA-Interim forcing data and performed all model coupling and model runs within the eWaterCycle platform (Hut et al., 2021). The eWaterCycle platform is a hydrological modeling platform that aims to improve the accessibility and reproducibility of hydrological models. On the eWaterCycle platform, hydrological models are run in

containers and 'communicate' with the central experiment that runs in a Jupyter Notebook. Communication with hydrological models is independent of the model language through GRPC4BMI (van den Oord et al., 2019) and BMI (Hutton et al., 2020).



Additionally, the ESMValTool (Eyring et al., 2016) implementation in eWaterCycle allows for smooth preprocessing and high compatibility of forcing data.

## 2.4 Basin runoff observations

Runoff observation data were obtained through the Global Runoff Data Centre (GRDC, 2020) for all basins except for the Rhone, for which we used observations from the French national hydrological service (Hydrobanque, 2020). Out of the 56 basins used by Huss and Hock (2018), 30 are present in the GRDC database with more than five years of daily runoff observations between 2000 and 2012. If a basin contained more than one gauging station in the GRDC database, we automatically selected the most upstream station that still included all the basin's glacier runoff, hereafter called the glacier sink (e.g. for

the Rhine, the gauging station in Basel was chosen instead of the most downstream station at Lobith). The glacier sinks were found using HydroSHEDS (Lehner et al., 2008). If the only available station was upstream of the glacier sink, we excluded the glaciers downstream of that station from our analysis. While the GRDC database does contain stations along the Rhone in Switzerland, the glacier sink is near the river mouth at Beaucaire. Therefore, observations at Beaucaire from the Hydrobanque were used as an alternative. The Supplementary material contains more information on the GRDC station numbers and the

available years, as well as a detailed map of all basins, their glacier coverage and the location of the gauging stations.

Of the 30 resulting basins, 5 were discarded from analysis for various issues related to river routing (see Supplementary material section 5). The remaining 25 large-scale glacierized basins are mostly concentrated in North-West America and Europe (Fig. 1). Openly available runoff data from rivers originating in the Himalayas are scarce, despite many of them being some of the world's most important and vulnerable glacier-fed river basins (Immerzeel et al., 2020). (Seasonally) arid regions

are likewise underrepresented, the only exceptions being the Rhone and the Negro river (respectively Cfb/Csa and Csb/Bsk on the Köppen-Geiger climate classification scale (Kottek et al., 2006)). On a practical note, since the vast majority of basins are located in the Northern Hemisphere, we will only mention the Northern Hemisphere months in the remainder of this work. The Southern Hemisphere equivalents will be implied for the Amazon, Negro and Clutha basins.

## 3 Methods

### 3.1 Model coupling

Within the context of this study, the term 'coupling' refers to the replacement of the PCR-GLOBWB 2 runoff by the GloGEM runoff for glacierized areas. To the best of our knowledge, this way of coupling has not been applied before for glaciohydrological modeling purposes. Several situations can be thought of for which further coupling between a glacier model and a hydrological model could be applied, such as surging glaciers damming upstream rivers (Sevestre and Benn, 2015) or the

flow of subglacial groundwater (Vincent et al., 2019), but these are considered irrelevant at the considered scale.

To ignore the PCR-GLOBWB 2 runoff originating from glacierized areas, we removed the fraction of the PCR-GLOBWB 2 landcover that corresponds to the glacierized area of the Randolph glacier inventory. This fraction is calculated per grid cell



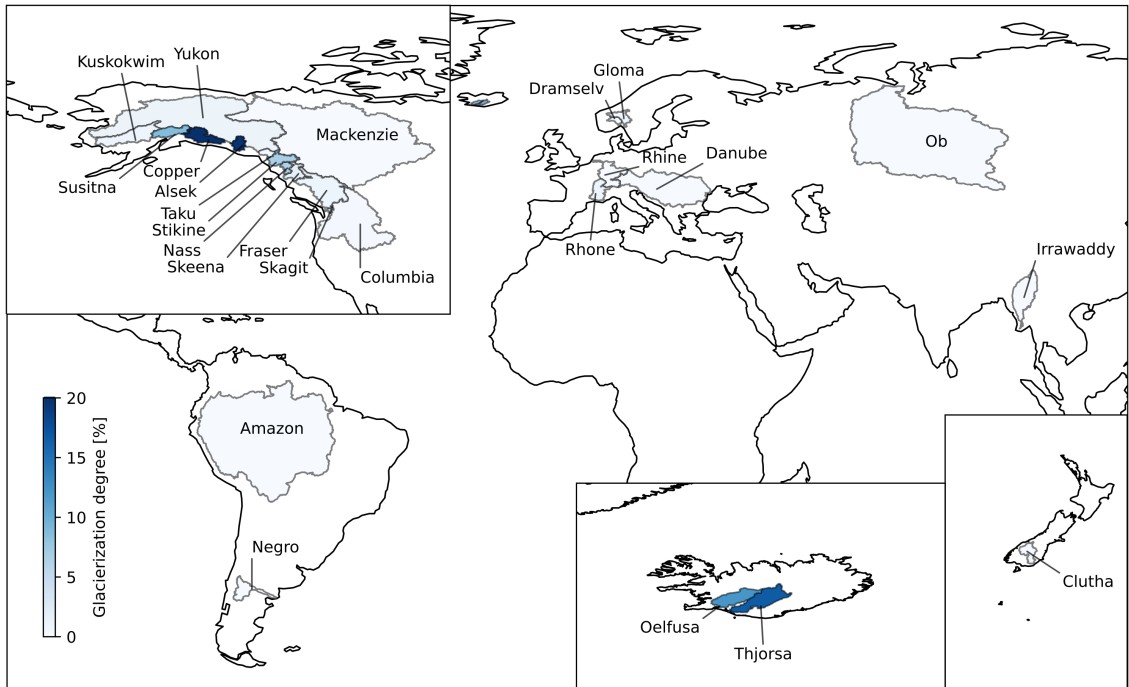

**Figure 1.** The 25 large-scale (>50.000 km$^2$) glacierized basins for which sufficient runoff observations are found. The hue represents the fraction of basin area covered by glaciers.

and subtracted from the short natural vegetation landcover class, since it is in the vast majority of cases the only landcover class present in glacierized grid cells. This operation prevents the PCR-GLOBWB 2 landcover classes from adding up to 1:

$$(f_{short\ natural\ veg.} - \mathbf{f_{glacier}}) + f_{tall\ natural\ veg.} + f_{paddy\ crop} + f_{non-paddy\ crop} = 1 - \mathbf{f_{glacier}} \tag{3}$$

Effectively, this causes PCR-GLOBWB 2 to omit any calculations on the glacier-covered area, without having to adjust the source code or forcing and without having to create a new landcover class. By not changing the source code, the reproducibility of this approach with other GHMs is increased. The only additional adjustment to be made was the disabling of the PCR-GLOBWB 2 setting that ensures the sum of landcover classes to be one.

As for the coupling itself, for each time step the GloGEM glacier runoff was added to the PCR-GLOBWB 2 variable *channel_storage*, which is equivalent to a direct routing into the stream. This is a simplification, since both under the glacier and below the glacier terminus groundwater infiltration is possible under certain conditions (Vincent et al., 2019; Castellazzi et al., 2019). Nevertheless, given the large scope of this study and the lacking research on this topic (Vincent et al., 2019) we ignored glacial groundwater recharge.





## 3.2 Model setups

Three different model setups were used. The *benchmark* is the default PCR-GLOBWB 2 model. The *coupled model* omits the PCR-GLOBWB 2 glacierized area and applies the GloGEM coupling instead (as discussed in section 3.1). Finally, the *bare model* is an auxiliary model setup that omits the PCR-GLOBWB 2 glacierized area but does not apply the GloGEM coupling. In theory, the difference between the benchmark and the bare model results in the routed PCR-GLOBWB 2 runoff for glacier-covered areas, while the difference between the coupled and the bare model is equal to the routed GloGEM runoff. We assumed the bare model to not include any glacier runoff. To initialize the models, one year with the climatological average of the period 1990-1999 was repeated 50 times as a model spin-up (Sutanudjaja et al., 2018).

## 3.3 Spilling prevention

Through the conversion of the basin boundaries from vector to raster format, a considerable part of the glaciers ended up in grid cells that were at the risk of being routed into adjacent basins, causing the runoff of these glaciers to be 'spilled'. To neutralize this spilling, the runoff of these glaciers was transferred to downstream grid cells that did not intersect with the basin vector boundary. . This spilling prevention was only applied to the coupled model and not to the benchmark, effectively leading to a larger total basin area for the coupled model (0-2% larger across all basins).

## 3.4 Evaluation metrics

### 3.4.1 Evaluation against the benchmark

To identify differences in basin runoff between the coupled model and the benchmark as a function of the time of the year, we applied the following normalized difference metric over all 25 basins:

$$ND_{t=d} = \frac{\overline{(Q_{Coupled} - Q_{Benchmark})}_{t=d}}{q_{99}(Q_{Coupled} - Q_{Benchmark})} \tag{4}$$

in which $ND$ stands for the Normalized Difference, $d$ is the calendar day, $Q$ is the basin runoff (in m$^3$ s$^{-1}$) and $q_{99}$ is the $99^{th}$ percentile of the difference taken over the whole time range. The normalization was applied with the $99^{th}$ percentile instead of the maximum difference to avoid the influence of extreme maxima. With this metric, a positive value indicates that on average the coupled model produces higher discharge than the benchmark on that particular calendar day, and vice versa.

### 3.4.2 Evaluation against observations

In the evaluation against the basin runoff observations, the difference in performance between the coupled model and the benchmark should be expressed relative to the highest possible performance difference (Seibert et al., 2018). After all, the same absolute error difference has larger implications on a day with little glacier melt than at the peak of the melt season. Since in this study the difference between the two models can only be attributed to a difference in glacier representation, we took the difference between zero glacier runoff and the maximum glacier runoff among PCR-GLOBWB 2 and GloGEM as





the maximum possible performance difference. This corresponds to the maximum difference among PCR-GLOBWB 2 and

GloGEM with the bare model (see section 3.2). The performance difference between the coupled model and the benchmark can then be expressed relative to the maximum possible performance difference as:

$$RRD = \frac{RMSE\left(Q_{Obs}, Q_{Coupled}\right) - RMSE\left(Q_{Obs}, Q_{Benchmark}\right)}{RMSE\left(Q_{Bare}, max(Q_{Coupled}, Q_{Benchmark})\right)} \tag{5}$$

in which $RMSE$ entails the root of the mean squared error and $RRD$ is the relative RMSE difference. With the $RRD$, a positive sign indicates whether the coupled model performs better compared to the benchmark or vice versa, while the value

indicates the fraction of the difference to the maximum possible difference. The RRD is therefore always between -1 and 1. A further justification of the metric choice is given in section 4 of the supplementary material.

### 3.4.3   Glacier influence metric

While the glacierization degree can give some indication of the hydrological importance of the glaciers in a river basin (Zhang et al., 2016; He et al., 2021), we used the results of the coupled model to formulate a direct measure of the importance of glacier

runoff on the basin scale. It is defined as the 99th percentile of the routed GloGEM runoff contribution to the coupled model daily runoff (GC99). The 99th percentile is chosen to reflect the crucial role of glaciers under extreme droughts (Huss, 2011). The threshold to distinguish weakly glacier-influenced basins from strongly glacier-influenced basins is chosen arbitrarily at GC99=0.5. In other words, in strongly glacier-influenced basins glacier runoff makes up more than 50% of the total basin runoff in 1% of the days. Out of 25 basins, this gives 9 strongly glacier-influenced basins between Susitna (GC=0.50) and Oelfusa

(GC99=0.85), and 16 weakly glacier-influenced basins between Amazon (GC99=0.003) and Kuskokwim (GC99=0.35).

## 4   Results

### 4.1   Hydrograph analysis

While 12 years of basin runoff are simulated for all 25 basins, we constrict the hydrograph analysis in this section to one representative year for 6 basins covering the full range of glacierization (Fig. 2). The complete collection of hydrographs is

presented in section 8 of the Supplementary material.

In weakly glacier-influenced basins (Mackenzie, Rhine, Columbia) the benchmark and the coupled model produce nearly indistinguishable results at the basin scale. This is also the case for the Rhone basin, where the strong glacier influence is only manifested in dry summers (e.g. 2003). The remaining strongly glacier-influenced basins (Alsek, Oelfusa) reveal that the coupled model produces higher runoff than the benchmark during the melt season. Compared to the runoff observations, the

benchmark has the tendency to overestimate the melt season runoff in weakly glacier-influenced basins, and vice versa for strongly glacier-influenced basins. Finally, in certain basins (e.g. Oelfusa, Columbia) the difference between the benchmark and the bare model is minimal, meaning that PCR-GLOBWB 2 generates virtually no runoff from glacierized areas in these basins.



The result of the resampling of the GloGEM glacier runoff from monthly to daily resolution (see section 2.2) is shown on the inverted axis of Fig. 2. Within each month, the daily glacier runoff fluctuations are deemed realistic, but between months sudden and rather unrealistic variations are visible (e.g. May to June for Mackenzie ). These variations are a consequence of the resampling having been performed for each month independently. A higher weighting factor $\alpha$ could potentially increase the sensitivity of the resampling to temperature and smooth out the jumps, although a sensitivity analysis demonstrates that this artefact does not significantly influence the runoff results (see section 6 of the Supplementary material).

### 4.2 Evaluation against the benchmark

The coupled model produces higher basin runoff than the benchmark for all basins throughout the melting season (Fig. 3a). The $ND$ shows a general pattern throughout the year for most basins with an increase from May to July, a peak in August and a decrease in September and October. Only a few weakly glacier-influenced basins (i.e. Amazon, Ob and Negro) deviate from this pattern. However, some basins (Fraser, Susitna, Kuskokwim) show slightly negative $ND$-values in May and October, indicating that here the coupled model temporarily produces lower runoff than the benchmark.

While the general $ND$ pattern is shared by nearly all basins, the impact this difference has on the total simulated runoff is greater in strongly glacier-influenced basins (Fig. 3b). In the Amazon, the coupled model runoff at the peak of the melt season (July and August in N.H.) is only 0.07% higher than the benchmark runoff, while in the Oelfusa this difference in peak runoff exceeds 250%.

### 4.3 Evaluation against observations

Over all basins, the coupled model performs worse at matching the observations than the benchmark. This is indicated by the mostly negative $RRD$-scores (57% $RRD$-scores<0, Fig. 4). However, when only considering the nine strongly glacier-influenced basins (GC99>0.5) the coupled model performs better (25/45 $RRD$-scores>0). This is particularly the case in July and August, at the peak of the ice melt season (14/18 $RRD$-scores>0). Furthermore, the average performance difference varies per month. Compared to the benchmark, the coupled model performs best in May (14/25 $RRD$-scores>0) and worst in September (5/25 $RRD$-scores>0). The coefficient of correlation ($R^2$) suggests a weak correlation of $RRD$-scores with glacier contribution for July and August, but no correlation for the other months. Note that the highest performance gain for the coupled model is achieved at the basin with the strongest glacier influence (Oelfusa, 5/5 $RRD$-scores>0).

A stand-alone performance evaluation of PCR-GLOBWB 2 is presented in Supplementary figure S2, showing positive Nash-Sutcliffe efficiency values (Nash and Sutcliffe, 1970) for basins with seasonal runoff regimes but negative calendar day benchmark efficiency values (Schaefli and Gupta, 2007) for all basins except the Rhone. Furthermore, the results of three alternatives to the $RRD$-metric are given in section 4 of the supplementary material, along with an explanation on why these metrics were not deemed suitable for this particular study.



**Figure 2.** Modelled and observed runoff of a representative selection of the 25 basins for a year close to average conditions. The left y-axis represents the runoff at the selected gauging station, while the right y-axis represents the GloGEM total basin glacier runoff. Note the different extents per basin on the y-axes. The 99th percentile of the GloGEM contribution to the coupled model daily runoff is presented with the basin name. The remaining hydrographs are presented in section 8 of the Supplementary material.





**Figure 3.** a) Mean normalized difference ($ND$) between the coupled model (PCR GLOBWB 2 & GloGEM) and the benchmark (only PCR-GLOBWB 2) for all 25 basins, showing that the coupled model produces higher runoff estimates throughout the melt season. The normalization is performed against the 99[th] percentile of the difference over the whole time range (2001-2012). The mean is computed for each calendar day over the same period. The solid black and dashed red lines represent the quartiles among the 25 basins. b) Ratio of the coupled model to the benchmark, averaged per month and over the period 2001-2012. The blue hue in both figures represents the 99[th] percentile of the routed GloGEM glacier runoff contribution to the coupled model runoff (GC99). The data of the three Southern Hemisphere basins are shifted six months forward in time to match the Northern Hemisphere months on the x-axis.




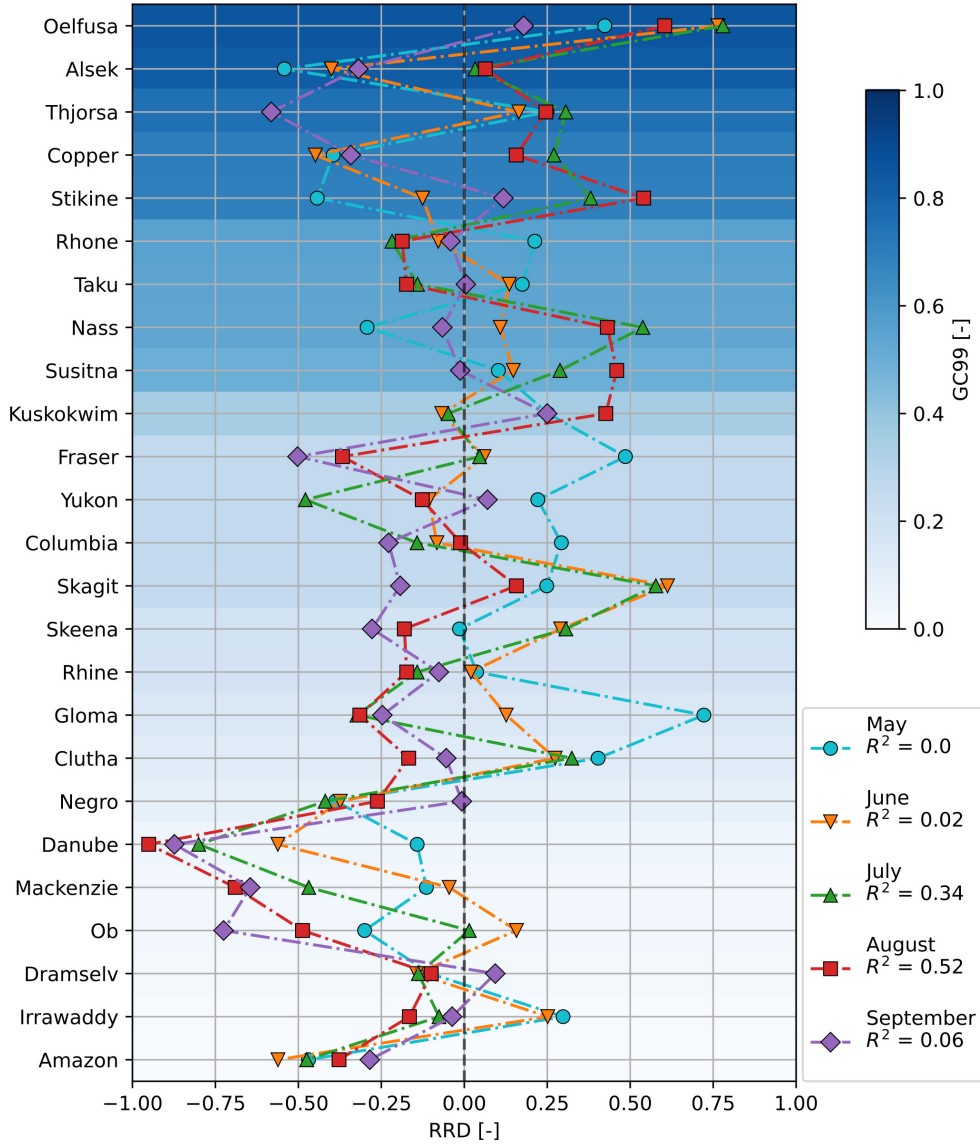

**Figure 4.** The Relative RMSE Difference ($RRD$) over all 25 basins throughout the melt season. The RMSE difference is calculated relative to the routed glacier runoff, which embodies the maximum possible RMSE difference. Positive scores indicate an improvement of the coupled model over the benchmark compared to observed runoff and vice versa. The $RRD$ always lies between −1 and 1. The basins are sorted based on the 99th percentile of the contribution of the routed GloGEM glacier runoff to the coupled model runoff (GC99). The coefficients of determination ($R^2$) represent the correlation of $RRD$-scores with the glacier contribution. The months are given only for the Northern Hemisphere, but the results of the three Southern Hemisphere basins are shown for November-March. The distinction between strongly and weakly glacier-influenced basins is set between Sustina and Kuskokwim (GC=0.5).



## 5    Discussion

### 5.1    Evaluation against the benchmark

#### 5.1.1    Overall difference

For four geographically representative basins we have identified several possible mechanisms to explain the overall runoff difference between the coupled model and the benchmark (Fig. 5). These mechanisms have been quantified on an annual basis to examine their contribution to the runoff difference. Since the coupling only applies to the glacier-covered area, the difference can only be attributed to the different representation of glaciers and the meteorological forcing.

Firstly, the lack of snow redistribution parameterization in PCR-GLOBWB 2 leads to the formation of 'snow towers' (Freudiger et al., 2017). Gravitational glacier flow, wind and avalanches are known to redistribute ice and snow from high elevations towards lower elevations, where melt is more likely to occur. Not accounting for these processes will lead to multiannual accumulation of snow at high elevations where temperatures rarely drop below melting point. As an extreme example, in an Amazonian glacierized grid cell this accumulation amounted up to 4 meter water equivalent per year. We should note that this snow accumulation is purely virtual and does not lead to an increase in the DEM. Out of the 25 basins, 17 simulate significant snow towers (see Supplementary figure S5). This phenomenon is acknowledged by Sutanudjaja et al. (2018) for polar regions. By contrast, GloGEM indirectly accounts for glacier flow through a geometry change module (Huss et al., 2010), which prevents the buildup of large amounts of static snow masses and ensures the transport of snow to lower elevations where melt is possible.

Secondly, PCR-GLOBWB 2 effectively treats glaciers as static rock masses and is therefore not able to capture changes in runoff following changes in glacier mass balance. Currently, many glaciers experience a peak in mass loss and therefore in glacier runoff (Huss and Hock, 2018), but in PCR-GLOBWB 2 no mass will be lost and therefore no additional inter-annual glacier runoff will be simulated. This problem was also noted by Sutanudjaja et al. (2018) after observing a negative correlation of the simulated total water storage with gravimetry measurements in Alaskan and Icelandic basins. Meanwhile, GloGEM was specifically developed to model global glacier mass balances and their dependence on future dynamic changes in ice extent. It has been calibrated to and validated against observations of multiple sources (Gardner et al., 2013; Hugonnet et al., 2021; WGMS, 2021) and is therefore likely to provide more reliable estimates of mass change-induced glacier runoff.

Thirdly, while the spilling prevention (section 3.3) may have helped in accurately routing all GloGEM glacier runoff in the coupled model, no similar measures were taken for the benchmark. Effectively this leads to a larger basin area for the coupled model (0-2% larger) and consequently to a greater basin runoff. This effect is greater in basins where a large portion of the glaciers is located at the basin boundary. The difference in basin runoff has been calculated for the four representative basins by performing additional coupled model runs without the spilling prevention and consequently calculating the difference with the original coupled model runs. While this difference ranged only between 1.5 and 4% for these four basins during the melt season, it could explain as much as 22% of the annual difference between the coupled model and the benchmark (see Columbia 2010 in Fig. 5).





The above-mentioned factors do not explain the entirety of the runoff difference between the coupled model and the benchmark. Particularly in the strongly glacier-influenced Alsek and Oelfusa basins large gaps are left unaccounted for by the
explanations above (i.e. the white space under the black lines in figures 5a and 5c). Evaporation/sublimation and groundwater recharge calculations are included in PCR-GLOBWB 2 and not in GloGEM. They could therefore (temporarily) account for part of the runoff difference, but their overall effect is estimated to be small. The differences can most likely almost entirely be attributed to precipitation correction factor $c_{prec}$ used in GloGEM. This correction factor allows for a scaling of the re-analysis grid cell precipitation to actual accumulation on the glacier. In fact, the elevation range occupied by glaciers is always strongly
underrepresented in the smoothed topography of the re-analysis, leading to an underestimation of orographically enhanced precipitation in the re-analysis product that needs to be accounted for (Immerzeel et al., 2015). Typically, re-analysis precipitation is upscaled by a factor of between 1.5 and 2.5 in order to correctly represent the observed mass flux components on glaciers (Huss and Hock, 2015; WGMS, 2021). Consequently, a higher grid cell precipitation in the coupled model is likely when it is applied without a counter-correction in PCR-GLOBWB 2. Unless the snow towers absorb this excess precipitation, it will
cause a higher runoff estimate than in the benchmark (albeit with a certain time lag).

### 5.1.2 Late spring difference

Despite the above-mentioned mechanisms causing PCR-GLOBWB 2 to underestimate the glacier runoff, there is nonetheless a short period in late spring where the benchmark still produces slightly higher runoff than the coupled model in many basins (late May in Fig. 3a). These basins only partially overlap with the basins in which no snow towers were found, and equally
no correlation with geographical location or climate was discovered. We hypothesize this effect to be the result of the limited horizontal and vertical spatial resolution of the temperature forcing in PCR-GLOBWB (Beek et al., 2011). Mountainous regions are characterized by steep horizontal and vertical temperature gradients, causing snow and glacier processes to be highly spatially dependent. If a model fails to capture these gradients due to an insufficient spatial or elevational resolution, there is a high chance of the melt being simulated too suddenly (Sexstone et al., 2020; Immerzeel et al., 2014). PCR-GLOBWB 2 does
facilitate a temperature downscaling from the 45 arcmin of ERA-Interim to the 5 arcmin model resolution using lapse rates from the CRU CL 2.0 climatology (New et al., 2002) to better account for snow dynamics, but this is arguably still too coarse for the gradients present at glacierized mountain areas. Beek et al. (2011) additionally hypothesize part of the melt timing error of PCR-GLOBWB 2 to be a consequence of the use of a constant melt rate and threshold temperature in the snow module. However, it should be mentioned that the high spatial resolution needed in mountainous areas is still rather unfeasible for
models that are designed to operate on a global scale, and therefore a certain degree of simplification will always be present. One solution to partly overcome this problem would be the use of a multi-resolution grid (e.g. Marsh et al. (2018); Özgen Xian et al. (2020). GloGEM does not downscale ERA-Interim data spatially, but it applies a set of twelve constant monthly temperature lapse rates derived from mean temperature in different pressure levels of the reanalysis to all glacier elevation bands (Huss and Hock, 2015). By covering a wider elevation range, GloGEM is likely to ensure a more gradual melt process
along late spring, particularly in the high temperature gradients around the highest elevations. In the present study this is partly





**Figure 5.** Mechanisms explaining the runoff difference between the coupled model and the benchmark. The black line represents the difference in annual runoff sums. The grey stack represents the annual increase in snow water equivalent modeled by PCR-GLOBWB 2 due to the lack of snow redistribution parameterization. The blue stack represents the annual net mass loss from retreating glaciers as modelled by GloGEM. The red stack represents the annual glacier runoff that would have spilled into neighboring basins as a consequence of basin boundary rasterization. This contributes to the imbalance since the spilling prevention is not applied to the benchmark. Note that the dark blue parts represent annual GloGEM net mass gain.

counteracted by the monthly jumps in glacier runoff owing to the temporal downscaling strategy (see section 4.1), but this effect is deemed of minor importance.



## 5.2 Evaluation against observations

When using the $RRD$ to evaluate the performance of the coupled model for reproducing observed basin runoff, there are two
main reasons to attribute more importance to the results obtained for strongly glacier-influenced basins. Firstly, the quality of
the glacier representation has greater implications in strongly glacier-influenced basins compared to weakly glacier-influenced
basins, and is therefore better reflected by the $RRD$. Secondly, in many weakly glacier-influenced basins PCR-GLOBWB 2
mostly overestimates the basin runoff (e.g. Danube, Ob, Irrawaddy) even without considering any glacier runoff (i.e. the bare
model). Since the coupling of GloGEM generally leads to even higher runoff, the $RRD$ will be mostly negative in weakly
glacier-influenced basins, even in the hypothetical case that the glacier runoff is simulated perfectly with the coupled model.

    The majority of $RRD$ values for strongly glacier-influenced (GC99>0.5) basins are positive: the $RRD$ is positive for 5 out of
9 values in May and June, 7 out of 9 in July and August and 2 out of 9 in September (Northern Hemisphere). Thus, particularly
at the peak of the melt season (July and August), the coupled model overall performs better than the benchmark. The lesser
performance in September can partially be explained by PCR-GLOBWB 2 reproducing the observations more closely, causing
the addition of GloGEM to lead to an overestimation (e.g. Thjorsa, Alsek). The highest scores over all metrics are obtained by
the basin with the highest maximum glacier contribution, the Oelfusa basin in Iceland. This is mostly explained by the heavy
underestimation of the summer runoff by PCR-GLOBWB 2.

    Considering the greater significance and higher scores of the $RRD$ in strongly glacier-influenced basins, we can conclude
that there is a high likelihood that the coupled model provides a better representation of glacier runoff than the benchmark.
While in this study the coupling does not lead to better results for weakly glacier-influenced basins, it is probable that the
glacier parameterization has in fact improved the resulting runoff in these basins, at least close to the headwaters, but that this
is not visible in the results.

## 6 Conclusions

We coupled the global hydrological model PCR-GLOBWB 2 with the global glacier model GloGEM to investigate whether
this coupling can lead to better GHM glacier representation and runoff predictions in glacierized basins. The coupling was
performed by adding the rasterized and resampled GloGEM glacier runoff to the channel storage of the PCR-GLOBWB 2 grid
cells. To avoid double counting, in each grid cell a fraction equal to the glaciation degree was subtracted from the grassland
landcover type. Both the uncoupled benchmark and the coupled model were run for 25 large-scale (>50.000 km$^2$) glacierized
basins across multiple continents during the hydrological years 2001-2012. The results were evaluated both mutually and
against GRDC runoff observations. The main outcomes are the following:

- The coupled model produces higher runoff across all basins. In July and August, this difference ranges from below 0.1%
for weakly glacier-influenced basins to more than 250% for strongly glacier-influenced basins. The difference can mainly
be attributed to an underestimation of runoff by PCR-GLOBWB 2, which simulates the formation of permanent 'snow
towers' and does not account for the additional melt induced by the retreat of glaciers worldwide.

– Nonetheless, in some basins the coupled model produces lower runoff than the benchmark in late spring, when the
        benchmark is likely to simulate a more abrupt onset of the melt season due to a limited spatial resolution.

       – In strongly glacier-influenced basins, where the coupling has the largest impact, the coupled model produces largely
        positive results in the evaluation against basin runoff observations. For weakly glacier-influenced basins an inverse trend
        is often observed, which can be linked to the coupling generally exacerbating the overestimation of basin runoff by
PCR-GLOBWB 2.

Combined, these outcomes suggest that the coupling of a global hydrological model and a global glacier model can lead to
a better representation of glaciers and, hence, high-mountain hydrology, and a high likelihood of increased runoff prediction
quality in glacierized basins. This study underlines the importance of glacier representation in strongly glacier-influenced
basins. Furthermore, it validates the feasibility of eWaterCycle II as a platform for hydrological modeling and model coupling.
Given the increased viability of global hydrological models in recent years and their nonetheless limited glacier represen-
tation, there is a large potential for future research. To further test the methodology of coupling a global hydrological model
with a global glacier model, future studies could apply ensembles of global hydrological and/or global glacier model, include
more basins around High Mountain Asia and/or perform a joint calibration. To facilitate such future work, we encourage fu-
ture global glacier model studies to include runoff estimates in the publication of results. Alternatively, to improve the glacier
representation within global hydrological models themselves, their developers could apply a multi-resolution grid and include
glacier mass balance estimates and basic glacier dynamics. Ultimately, an improved glacier representation in GHMs could lead
to a better understanding of the global patterns of present and future hydrology of large-scale glacierized basins.

*Code availability.* All code and supporting files used in this study is available at https://doi.org/10.5281/zenodo.6386306.

*Author contributions.* Conceptualization: PW, RH & JA
Data curation, formal analysis, project administration, visualization & writing-original draft preparation: PW
Methodology: PW, MHu, ES
Resources and software: ND, JA, RH
Supervision & validation: RH, JA, HZ, MHr
Writing - review & editing: all authors

*Competing interests.* Co-author Markus Hrachowitz is a member of the editorial board of the HESS journal.

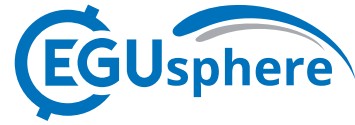

*Acknowledgements.* Harry Zekollari acknowledges the funding received from a EU Horizon 2020 Marie Skłodowska-Curie Individual Fellowship (grant no. 799904) and from the Fonds de la Recherche Scientifique – FNRS (postdoctoral grant – chargé de recherches). Jerom Aerts and Rolf Hut have received funding from the Netherlands eScience Center under file number 027.017.F01.



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
