# Peer review of "Coupling a global glacier model to a global hydrological model prevents underestimation of glacier runoff"

_EGUsphere, 2022_

## Author Comment (AC1)

**Reply to reviewer #1**

*The manuscript tests the hypothesis that coupling a global glacier model (i.e., GloGEM) with a global hydrological model (i.e., PCR-GLOBWB 2) leads to a more realistic glacier representation and improved runoff prediction in the global hydrological model. both the uncoupled benchmark and the coupled model were run for 25 large-scale glacierized basin during the hydrological years 2001-2012. Overall, the manuscript is clearly written, and the results are well discussed.*

We highly appreciate this positive overall assessment of our manuscript.

*I have only two main concerns and one specific point for this study:*

- *The authors test a widely accepted hypothesis that the physical representation and simulation of hydrological model will be improved if its corresponding parameterization is optimized on a global scale. I am not quite sure that a test of a widely accepted hypothesis is a true innovation (I leave this question to the editor). If the test is done by coupling the global hydrological model and global glacier model physically instead of simply replacing the PCR-GLOBWB 2 runoff by the GloGEM runoff for glacierized areas, the novelty of this study make sense at least from a practical point of view.*

We acknowledge the reviewer for expressing this valid concern. We argue that replacing the PCR-GLOBWB 2 runoff by the GloGEM runoff for glacierized areas is a physical coupling that goes beyond parameterization of the hydrological model. We structure our reply into two parts:

- Glaciers lie at the origin of streams were the exchange of water between glaciers and the rest of the catchment predominantly takes place on the surface level and in only one direction. Replacing the glacier runoff of one model with the glacier runoff of another model as a coupling method does therefore not produce large physical inconsistencies. One simplification we applied is that all glacier runoff ends up in the stream and that therefore none of it drains to the subsurface before reaching the stream. Although there is exchange in the subsurface we consider this simplification acceptable at the global scale as we describe in section 3.1 (lines 136 – 140). This simplification makes the coupling of a glacier model with a hydrological model feasible and straightforward to implement without losing too much physical basis. Such a straightforward implementation of one-way coupling of models demonstrates that this method can be adopted by other global hydrological models. The slight loss in physical basis induced by the coupling method can then be compensated by the gain in glacier representation accuracy of the GGM relative to the GHM.
- Global hydrological models without a sufficiently detailed glacier representation will likely produce less reliable results in glacierized basins. All new methodologies that have the potential to alleviate this problem, among which the one presented in this study, are an improvement of the current state of research in hydrology. Additionally, this study could also incentivize GHM model developers to carefully evaluate and improve the physical process descriptions applied in their respective GHMs in the future. The relevance of this new methodology is exemplified by the rise in GHM-related publications, the large amount of people depending on glacial runoff worldwide and the significant projected climate change induced changes in glacier runoff.

We will adjust the first paragraph of section 3.1 to the following: "Within the context of this study, the term 'coupling' refers to the replacement of the PCR-GLOBWB 2 runoff by the GloGEM runoff for glacierized areas. **We deem this simplification of coupling physically plausible since much of the**

**exchange of water between glaciers and the rest of the catchment occurs at the surface in the form of runoff.** To the best of our knowledge, this **coupling approach** has not been applied before for glacio-hydrological modeling purposes. Several situations can be thought of for which further coupling between a glacier model and a hydrological model could be applied, such as surging glaciers damming upstream rivers (Sevestre and Benn, 2015) or the flow of subglacial groundwater (Vincent et al., 2019), but these are considered irrelevant at the considered scale."

- *In both Abstract and Introduction sections, the authors mentioned that global runoff prediction can be improved through the coupling of GHMs and GGMs. However, only runoff "simulation" was tested in this study rather than "prediction". The authors are suggested showing the results of runoff prediction (not for the calibration/validation periods but for the prediction period) as well.*

We thank the reviewer for this comment. The reviewer is correct in pointing out that the use of the term "prediction" is not accurate and therefore we will adjust it in the revised manuscript. As the reviewer states correctly, we only performed historical simulations and no predictions. However, the simulations can be seen as a test of the prediction quality, since PCR-GLOBWB 2 is not calibrated over the simulation years. The results show that the current hydrological model carries large uncertainties in the simulation of glacierized basins, also after the coupling. Therefore, we deem it unsuitable to perform future simulations with in glacierized catchments and this is left for future studies.

To clarify this part, we will add a sentence at the end of section 3.2: "All model setups are run between the hydrological years 2000-2012. As PCR-GLOBWB 2 is not calibrated, the simulation over these 12 years can be seen as a test for the prediction quality of the coupled model versus the uncoupled model."

*A specific point: Paragraph 165 in P7, extra periods.*

Noted and will be adjusted.

---

## Author Comment (AC2)

**Reply to reviewer #2**

*In general the manuscript is quite interesting and the contribution of a global glacier model coupled with a global hydrologic hydrologic model certainly has the potential to be an important and significant contribution to our field.*

We appreciate this affirmation of the relevance of our study.

*I do have a few but important major comments that relate to both the model coupling and verification of the process representation.*

*(1) It would be useful to speak more about the coupling approach and its compatibility with BMI (L110-113). In particular, this sentence needs to be expanded on: "Communication with hydrological models is independent of the model language through GRPC4BMI (van den Oord et al., 2019) and BMI (Hutton et al., 2020). Additionally, the ESMValTool (Eyring et al., 2016) implementation in eWaterCycle allows for smooth preprocessing and high compatibility of forcing data."*

*Firstly, please spell out BMI in its first use. In the US in particular, BMI is rapidly becoming the standard for model coupling. It should be noted as to how the coupling approach here is compliant with the BMI standard or can be adapted to BMI.*

*If the code is not usable with the BMI standard, please add detail to the text so that a user understands that (at least from what is implied by the above sentence) they can use something other than a Jupyter Notebook (Python, for example) to couple the models - because of the language-independent nature of the coupling used by the different modeling components.*

*Hopefully, this is the case, as it is certainly an important advancement beyond the potential scientific improvements offered with respect to the process representation.*

We agree with the reviewer that we should further elaborate on what BMI is, how we use it for model coupling, and which parts of the coupling are done with or without the BMI interface. In section 2.3 (L110-113), we only discuss the use of BMI on the eWaterCycle platform itself, but not whether the coupling is also established using BMI. Below we explain this in detail and suggest changes to the manuscript.

As discussed in section 3.1 (L141-L154), the numerical implementation of the coupling consists of two steps: the addition of the glacier runoff calculated from the Global Glacier Evolution Model (GloGEM) and the removal of the PCR-GLOBWB 2 glacier runoff. As for the first step, the *channel_storage* variable in PCR-GLOBWB 2 can be adjusted using BMI, and thus the GloGEM glacier runoff is added to PCR-GLOWB 2 using the get_value() and set_value() BMI functions. This communication is established using the existing eWaterCycle architecture: BMI calls from the central python notebook (client) are conveyed and translated to the BMI implementation in PCR-GLOBWB 2 (server) using GRPC4BMI, and vice versa. Concerning the second step, the adjusted landcover fraction maps (which exclude all glacierized area) can be handed over to the model using a class in the eWaterCycle package that creates the model configuration file, which is then passed to the model via the BMI initialize function. This communication therefore also takes place through a standard model interface and not at the model itself. The creation of the adjusted landcover fraction maps needs to be done manually however. The flowchart below illustrates this numerical implementation of the coupling, and the code snippets show how the BMI and eWaterCycle functions (in yellow) are used for the coupling of GloGEM (Github link).

[Figure]

```
219  if ADJUST_LANDCOVER!=False:
220      if ADJUST_LANDCOVER=='GRASSLANDS':
221          FRAC_PREFIX = 'grassf'
222      elif ADJUST_LANDCOVER=='PROPORTIONAL':
223          FRAC_PREFIX = 'propf'
224      else:
225          print('Wrong input ADJUST_LANDCOVER')
226      parameter_set.config['landSurfaceOptions']['noLandCoverFractionCorrection']='True'
227      parameter_set.config['forestOptions']['fracVegCover']    ='glaciers_landcov/'+FRAC_PREFIX+'_tall.map'
228      parameter_set.config['grasslandOptions']['fracVegCover']='glaciers_landcov/'+FRAC_PREFIX+'_short.map'
229      parameter_set.config['irrPaddyOptions']['fracVegCover'] ='glaciers_landcov/'+FRAC_PREFIX+'_pad.map'
230      parameter_set.config['irrNonPaddyOptions']['fracVegCover']='glaciers_landcov/'+FRAC_PREFIX+'_nonpad.map'
231  elif ADJUST_LANDCOVER==False:
232      parameter_set.config['landSurfaceOptions']['noLandCoverFractionCorrection']='False'
233      parameter_set.config['forestOptions']['fracVegCover']    ='global_05min/landSurface/landCover/naturalTall/vegf_tall.map'
234      parameter_set.config['grasslandOptions']['fracVegCover']='global_05min/landSurface/landCover/naturalShort/vegf_short.map'
235      parameter_set.config['irrPaddyOptions']['fracVegCover'] ='global_05min/landSurface/landCover/irrPaddy/fractionPaddy.map'
236      parameter_set.config['irrNonPaddyOptions']['fracVegCover']='global_05min/landSurface/landCover/irrNonPaddy/fractionNonPaddy.map'
237  else:
238      print('Wrong input ADJUST_LANDCOVER')
239
240  parameter_set.save_config(glob_ini_new)
```

```python
print('call docker')
if CLUSTER == True:
    from grpc4bmi.bmi_client_singularity import BmiClientSingularity
    pcrg = BmiClientSingularity(image='ewatercycle-pcrg-grpc4bmi.sif',
                                input_dir=RUN_DIR,
                        output_dir=OUT_DIR)
else:
    from grpc4bmi.bmi_client_docker import BmiClientDocker
    pcrg = BmiClientDocker(image='ewatercycle/pcrg-grpc4bmi:setters', image_port=55555,
                    input_dir=RUN_DIR,
                    output_dir=OUT_DIR)

print('input: '+pcrg.input_dir)
print('output: '+pcrg.output_dir)
**initialize**
start_time = time.time()
time.sleep(18)
print ('Initialize...')
pcrg.initialize(glob_ini_new)
**%% Start Run**
i=0
start_time = time.time()
while pcrg.get_current_time()!=pcrg.get_end_time(): #Testrun=True gives another end condition
    full_loop_timer = time.time()
    if COUPLE_GLOGEM:
        #Coupling is done by adding the glogem runoff to the channel_storage
        chan_stor    = pcrg.get_value('channel_storage') #Gives flat array
        chan_stor    = np.reshape(chan_stor,(LATSIZE,LONSIZE))

        # chan_stor[isglac]=nc.R.isel(time=i).data[isglac]

        if nc.lat.data[0]>nc.lat.data[-1]: #(if Southern Hemisphere)
            chan_stor  += nc.R.data[i,::-1,:]
        else:
            chan_stor    += nc.R.data[i,:,:]

        chan_stor    = chan_stor.flatten()

        ####
        pcrg.set_value('channel_storage',chan_stor)
        ####

        # Set values with flat indices
        #pcrg.set_value_at_indices('channel_storage',
        #                          np.flatnonzero(isglac.data),
        #                          nc.R.isel(time=i).data[isglac])

    pcrg_timer  = time.time()
    ##
    pcrg.update()
```

In conclusion, both the addition of the GloGEM runoff and the removal of the PCR-GLOBWB 2 runoff are established using a BMI. Only the creation of the adjusted landcover maps needs to be performed manually. We will make two adjustments to the manuscript to incorporate the reviewer's feedback:

- Swap BMI and GRPC4BMI and spell out BMI in L110, which thus becomes: "Communication with hydrological models is independent of the model language through the Basic Model Interface (BMI) (Hutton et al., 2020) and GRPC4BMI (van den Oord et al., 2019)."
- Adding a new paragraph after L154: "The numerical implementation of the coupling is largely done using standard BMI functionality. As mentioned in section 2.3, the eWaterCycle platform uses BMI for communication with the hydrological models and therefore also allows for requesting and modifying model variables using the *get_value()* and *set_value()* BMI functions. In this case, these functions are used to add the GloGEM glacier runoff to PCR-GLOBWB 2 but other combinations of glacier and hydrological models could be coupled using the same interface. While the adjusted landcover fraction maps need to be created manually, they are passed to the model via the model's configuration file in the BMI *initialize()* function.

*(2) In the results and conclusions (and abstract), (see L341 as an example), the statement is made that "The coupled model produces higher runoff across all basins." However, there is not a follow-on statement discussing whether this results in better hydrological modeling of the process in that it matches observations. This type of assessment should be more clearly stated in these sections. Simply producing more runoff through this new model coupling does not necessarily mean the modeling results are better or the process representation is more correct. Using an evidence-based approach with a comparison to observations, and showing how this is an improvement over other modeling approaches is preferred. Otherwise, the manuscript's hydrologic contribution is reduced and more emphasis is placed on the model coupling/software engineering contribution. I would be interested to understand the authors' response to this comment.*

If we understand this comment correctly, the reviewer here points to an identifiability problem in our study: when only relying on downstream runoff observations, it is difficult to tell whether (I) the actual glacier representation is improved and glacier runoff underestimation is prevented, or whether (ii) the GloGEM runoff simply compensates for other deficits at the basin level but is an actual overestimation of the glacier runoff on itself. The reviewer is correct in pointing this out, we acknowledge the existence of this problem and the fact that we did not elaborate on it sufficiently.

The best (and perhaps only) way to solve this problem would be to have access to isolated observations of each glacier's runoff, but this is clearly unfeasible considering the thousands of glaciers involved in this study. We did use the observations of one glacier's runoff (Greater Aletsch glacier, with a volume of ca. 15 $km^3$ the largest of the European Alps) to manually calibrate the temporal downscaling parameter. The results are shown in section two of the supplementary material, and they clearly indicate that at least for this particular glacier PCR-GLOBWB 2 heavily underestimates the glacier runoff with a constant value of zero.

[Figure]

 An additional aspect of this study that helps to solve the identifiability problem is the fact that GloGEM has been calibrated on and validated against glacier mass balance observations of different sources (Gardner2013, WGMS 2021). This means that at least on a monthly timescale, the GloGEM glacier runoff is unlikely to be heavily overestimated.

The identifiability problem can be further reduced by taking the runoff observations as close as possible to the glaciers to limit the influence of other runoff generation mechanisms. Therefore, the discharge stations chosen in this study were chosen as upstream as possible while still including all glacier runoff (L118-124). However, we did limit ourselves to a single discharge station per basin, even when other upstream discharge stations were available. Including these would have led to a further reduction of the identifiability problem and a more thorough evaluation of our results. This would be something to consider in future studies.

The final point that we believe provides most of the solution to the identifiability problem is the evidence of the glacier runoff underestimation of PCR-GLOBWB 2. We showed that snow towers are present in a majority of the basins (L252-261), that PCR-GLOBWB 2 does not include the additional glacier runoff from glacier mass loss (L74-76, 262-265), and that these two factors are responsible for a large portion of the difference between the benchmark and the coupled model (figure 5).

To try to take away the doubts that the reviewer had and that other readers might have, we will add a paragraph and adjust the final paragraph in section 5.2 as follows:

"A major limitation of using runoff observations at the basin outlet is that they are not a direct measure of glacier runoff, and therefore we can not fully exclude the possibility that GloGEM overestimates the glacier runoff and simply compensates for other deficits of PCR-GLOBWB 2 at the basin level to reach the higher RRD-scores. While we chose the discharge stations as close to the glacier sink as possible, we excluded in many cases other upstream discharge stations from our analysis. Future studies are encouraged to consider multiple discharge stations per basin to limit this identifiability problem. Nonetheless, several aspects of our study point against the abovementioned possibility. Firstly, since GloGEM has been calibrated and validated with glacier mass balance observations (Gardner et al., 2013) it is unlikely that GloGEM heavily underestimates glacier runoff, at least on a monthly scale. Secondly, an indication that the PCR-GLOBWB 2 underestimation stems from glacierized areas is given by the observation at the Aletsch glacier (see section 2 of the Supplement), where PCR-GLOBWB 2 simulates zero runoff over multiple years.  Finally, in section 5.1 we provide evidence that the difference in glacier parameterization between PCR-GLOBWB 2 and GloGEM is responsible for a large part of the difference in runoff.

In conclusion, strongly glacier-influenced basins produce at the same time higher and more significant RRD scores, and we have shown this to be mostly attributable to the difference in glacier representation. The coupling of GloGEM is therefore likely to prevent significant underestimation of

glacier runoff in PCR-GLOBWB 2. While in this study the coupling does not lead to better results for weakly glacier-influenced basins, it is probable that the glacier parameterization has in fact improved the resulting runoff in these basins, at least close to the headwaters, but that this is not visible in the results."

---

## Author Response (AR1)

Dear editor and reviewers,

Here we provide a point-by-point response of the changes we made based on the reviews. Thank you for the valuable feedback. We believe that this greatly benefitted the quality of the manuscript.

Kind regards on behalf of all authors,

Pau Wiersma

**Point by point response to Reviewer #1**

*I have only two main concerns and one specific point for this study:*

- *The authors test a widely accepted hypothesis that the physical representation and simulation of hydrological model will be improved if its corresponding parameterization is optimized on a global scale. I am not quite sure that a test of a widely accepted hypothesis is a true innovation (I leave this question to the editor). If the test is done by coupling the global hydrological model and global glacier model physically instead of simply replacing the PCR-GLOBWB 2 runoff by the GloGEM runoff for glacierized areas, the novelty of this study make sense at least from a practical point of view.*

We adjusted the first paragraph of section 3.1 to the following: "Within the context of this study, the term 'coupling' refers to the replacement of the PCR-GLOBWB 2 runoff by the GloGEM runoff for glacierized areas. **We deem this simplification of coupling physically plausible since much of the exchange of water between glaciers and the rest of the catchment occurs at the surface in the form of runoff.** To the best of our knowledge, this **coupling approach** has not been applied before for glacio-hydrological modeling purposes. Several situations can be thought of for which further coupling between a glacier model and a hydrological model could be applied, such as surging glaciers damming upstream rivers (Sevestre and Benn, 2015) or the flow of subglacial groundwater (Vincent et al., 2019), but these are considered irrelevant at the considered scale."

- *In both Abstract and Introduction sections, the authors mentioned that global runoff prediction can be improved through the coupling of GHMs and GGMs. However, only runoff "simulation" was tested in this study rather than "prediction". The authors are suggested showing the results of runoff prediction (not for the calibration/validation periods but for the prediction period) as well.*

We added a sentence at the end of section 3.2: "All model setups are run between the hydrological years 2000-2012. As PCR-GLOBWB 2 is not calibrated, the simulation over these 12 years can be seen as a test for the prediction quality of the coupled model versus the uncoupled model."

*A specific point: Paragraph 165 in P7, extra periods.*

This has been adjusted.

**Point by point response to Reviewer #2**

*I do have a few but important major comments that relate to both the model coupling and verification of the process representation.*

*(1) It would be useful to speak more about the coupling approach and its compatibility with BMI (L110-113). In particular, this sentence needs to be expanded on: "Communication with hydrological models is independent of the model language through GRPC4BMI (van den Oord et al., 2019) and BMI (Hutton et al., 2020). Additionally, the ESMValTool (Eyring et al., 2016) implementation in eWaterCycle allows for smooth preprocessing and high compatibility of forcing data."*

*Firstly, please spell out BMI in its first use. In the US in particular, BMI is rapidly becoming the standard for model coupling. It should be noted as to how the coupling approach here is compliant with the BMI standard or can be adapted to BMI.*

*If the code is not usable with the BMI standard, please add detail to the text so that a user understands that (at least from what is implied by the above sentence) they can use something other than a Jupyter Notebook (Python, for example) to couple the models - because of the language-independent nature of the coupling used by the different modeling components.*

*Hopefully, this is the case, as it is certainly an important advancement beyond the potential scientific improvements offered with respect to the process representation.*

We made two adjustments to the manuscript to incorporate the reviewer's feedback:

- Swap BMI and GRPC4BMI and spell out BMI in L110, which thus becomes: "Communication with hydrological models is independent of the model language through the Basic Model Interface (BMI) (Hutton et al., 2020) and GRPC4BMI (van den Oord et al., 2019)."
- Adding a new paragraph after L154: "The numerical implementation of the coupling is largely done using standard BMI functionality. As mentioned in section 2.3, the eWaterCycle platform uses BMI for communication with the hydrological models and therefore also allows for requesting and modifying model variables using the *get_value()* and *set_value()* BMI functions. In this case, these functions are used to add the GloGEM glacier runoff to PCR-GLOBWB 2 but other combinations of glacier and hydrological models could be coupled using the same interface. While the adjusted landcover fraction maps need to be created manually, they are passed to the model via the model's configuration file in the BMI *initialize()* function.

*(2) In the results and conclusions (and abstract), (see L341 as an example), the statement is made that "The coupled model produces higher runoff across all basins." However, there is not a follow-on statement discussing whether this results in better hydrological modeling of the process in that it matches observations. This type of assessment should be more clearly stated in these sections. Simply producing more runoff through this new model coupling does not necessarily mean the modeling results are better or the process representation is more correct. Using an evidence-based approach with a comparison to observations, and showing how this is an improvement over other modeling approaches is preferred. Otherwise, the manuscript's hydrologic contribution is reduced and more emphasis is placed on the model coupling/software engineering contribution. I would be interested to understand the authors' response to this comment.*

We added a paragraph and adjusted the final paragraph in section 5.2 as follows:

"A major limitation of using runoff observations at the basin outlet is that they are not a direct measure of glacier runoff, and therefore we can not fully exclude the possibility that GloGEM overestimates the glacier runoff and simply compensates for other deficits of PCR-GLOBWB 2 at the basin level to reach the higher RRD-scores. While we chose the discharge stations as close to the glacier sink as possible, we excluded in many cases other upstream discharge stations from our analysis. Future studies are encouraged to consider multiple discharge stations per basin to limit this identifiability problem. Nonetheless, several aspects of our study point against the abovementioned possibility. Firstly, since GloGEM has been calibrated and validated with glacier mass balance observations (Gardner et al., 2013) it is unlikely that GloGEM heavily underestimates glacier runoff, at least on a monthly scale. Secondly, an indication that the PCR-GLOBWB 2 underestimation stems from glacierized areas is given by the observation at the Aletsch glacier (see section 2 of the Supplement), where PCR-GLOBWB 2 simulates zero runoff over multiple years. Finally, in section 5.1 we provide evidence that the difference in glacier parameterization between PCR-GLOBWB 2 and GloGEM is responsible for a large part of the difference in runoff.

In conclusion, strongly glacier-influenced basins produce at the same time higher and more significant RRD scores, and we have shown this to be mostly attributable to the difference in glacier representation. The coupling of GloGEM is therefore likely to prevent significant underestimation of glacier runoff in PCR-GLOBWB 2. While in this study the coupling does not lead to better results for weakly glacier-influenced basins, it is probable that the glacier parameterization has in fact improved the resulting runoff in these basins, at least close to the headwaters, but that this is not visible in the results."

**Response to the editor's final comment**

*The first reviewer raised a question of novelty in the manuscript, which I believe was answered well by the authors but it would be helpful to add more of the response to the text. I would suggest that the last sentences of Bullet 2, Response to Comment 1 be added somewhere in the introduction so that the text could read, "We hypothesize that this simplification makes the coupling of a glacier model with a hydrological model feasible and straightforward to implement without losing too much physical basis. Such a straightforward implementation of one-way coupling of models could demonstrate that this method can be adopted by other global hydrological models. We further hypothesize that the slight loss in physical basis induced by the coupling method can then be compensated by the gain in glacier representation accuracy of the GGM relative to the GHM."*

We added two sentences to the end of the introduction: "To benefit its replicability with other GHMs, we apply a simplified coupling method using standard open source libraries. We expect the gain in glacier representation accuracy of the GGM relative to the GHM to compensate for any loss in physical basis following the simplifications applied in the coupling method."

**Response to external comments by Sarah Hanus**

We added several missing basins in Table S1 of the supplementary material. Additionally, we specified in figure S5 that the SWE in the figures includes both accumulated "glacier" and actual snow mass over the spinup period, that no importance should be attributed to the absolute values of SWE and that the SWE values only reflect the glacierized areas of the basins.

**Overview of changes**

- Introduction

- - Updated the Müller-Schmied et al. reference to refer to the published article instead of the preprint
    - Discussed the expected gain in physical basis
- Models and data
    - Updated the Hut et al. reference to refer to the published article instead of the preprint
    - Swapped the order of the BMI and GRPC4BMI references
- Methods
    - Further discussed the physical basis of the methodology
    - Discussed the nature of the BMI implementation in the coupling
    - Specified the distinction between prediction and simulation in this study
- Discussion
    - Added a paragraph in the section "evaluation against observations" on the identifiability problem and the argumentation on how we believe to have overcome it
- Supplementary material
    - Added missing basins in Table S1
    - Elaborated on the SWE interpretation in figure S5